# FEW-SHOT LEARNING WITH BIG PROTOTYPES

## ABSTRACT

Using dense vectors, i.e., prototypes, to represent abstract information of classes has become a common approach in low-data machine learning scenarios. Typically, prototypes are mean output embeddings over the instances for each class. In this case, prototypes have the same dimension of example embeddings, and such tensors could be regarded as "points" in the feature space from the geometrical perspective. But these points may lack the expressivity of the whole class-level information due to the biased sampling. In this paper, we propose to use tensor fields ("areas") to model prototypes to enhance the expressivity of class-level information. Specifically, we present *big prototypes*, where prototypes are represented by hyperspheres with dynamic sizes. A big prototype could be effectively modeled by two sets of learnable parameters, one is the center of the hypersphere, which is an embedding with the same dimension of training examples. The other is the radius of the sphere, which is a scalar. Compared with atactic manifolds with complex boundaries, representing hypersphere with parameters is immensely easier. Moreover, it is convenient to perform metric-based classification with big prototypes in few-shot learning, where we only need to calculate the distance from a data point to the surface of the hypersphere. Extensive experiments on few-shot learning tasks across NLP and CV demonstrate the effectiveness of big prototypes.

## 1 INTRODUCTION

Learning from few examples, i.e., few-shot learning, receives increasing attention in modern deep learning. On the one hand, constituting cognition of novel concepts with few instances is a crucial way for machines to imitate human intelligence. On the other hand, annotating large-scale supervised datasets is expensive and time-consuming (Lu et al., 2020). Although traditional deep neural models have achieved tremendous success under sufficient supervision, it is still challenging to produce comparable performance when training examples are limited. Hence, a series of studies are proposed to generalize deep neural networks to low-data scenarios. One crucial branch of them is meta-learning with prototypes (Reed, 1972; Nosofsky, 1986; Snell et al., 2017), where models are trained to quickly adapt to current tasks and carry out classification via metric-based comparisons between examples and newly introduced variables, prototypes of those classes.

In a general way, prototypes are designed to represent abstract class-level information and calculated by taking the mean output of a few examples belonging to one same class. Thus, prototypes are represented as dense vectors with the same dimensions as the embeddings of training examples and can be regarded as center points of those classes in the embedding space. And for the queried examples that the model needs to predict, the basic idea is to calculate the distances between the queried examples and prototypes and conduct classification based on the distances. Originated from Prototypical Network (Snell et al., 2017), sets of derivative prototype-based methods demonstrate the effectiveness in few-shot learning (Ren et al., 2018; Gao et al., 2019a; Allen et al., 2019; Pan et al., 2019; Ding et al., 2021a). However, prototypes are estimated from a few sampled examples, which cannot uncover the overall ground truth distribution (Yang et al., 2021). Such biased distributions may generate biased prototypes and trigger sequent classification errors. In other words, this modeling approach of prototypes could lack the capability to express the universal class-level information. Hence, to enhance such expressibility of prototypes, we propose to use areas, i.e., tensor fields, rather than points in the embedding space to represent the class-level information.

In this paper, we propose *big prototypes* to use hyperspheres in the feature space to abstract the class-level information, and the feature points can be distributed inside or around the big prototypes.

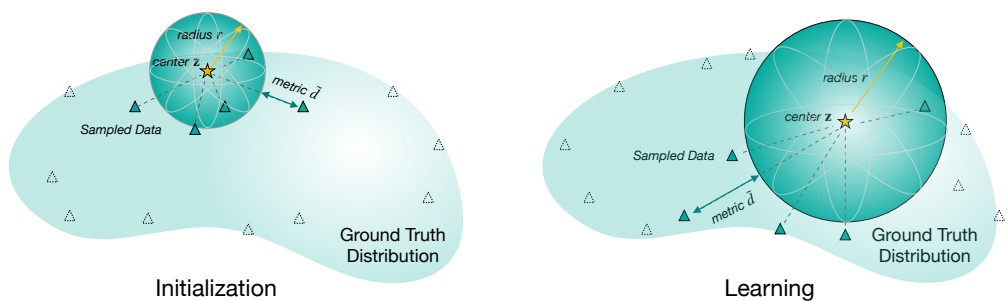

Figure 1: The illustration of our proposed *big prototypes*, where the data is sampled in 5-shot. The star symbol denotes the center of the hypershpere, the solid triangle denotes the sampled examples and the dotted triangle denotes other examples in the whole dataset. The green solid line denotes the distance from a data embedding to the surface of the hypersphere. The left part illustrates the initialization stage, where the center and radius are estimated by the sampled data, and the the right part illustrates the learning stage, where the center and radius are simultaneously optimized. We can see that the location of the center and the value of the radius may change during the learning process.

Such modeling is equipped with two obvious advantages: easy to model and easy to calculate the distances. On the one hand, even if we attempt to use areas but not points to represent class-level information, it is difficult to explicitly characterize manifolds with complex boundaries in deep learning. But via hyperspheres modeling, we can obtain a big prototype only through two sets of parameters: the center and the radius of hyperspheres. On the other hand, hyperspheres are suitable for calculating the distances in Euclidean space. We can simply calculate the distance from one feature point to the surface of the hypersphere to perform metric-based classification, which is also difficult for other manifolds. Moreover, it is easy to combine these two advantages in few-shot learning, the distances from one feature point to the surface of a big prototype can be formalized as the distance from the point to the center of the hypersphere minus the radius. Thus, both radius and hypersphere center can appear in the loss function and participate in the backward propagation during optimization. Intuitively, for the classes with sparse feature distribution, the corresponding radii of their prototypes are large, and the radii are small otherwise.

We conduct extensive experiments in both natural language processing (NLP) and computer vision (CV) to evaluate the effectiveness of big prototypes. Specifically for NLP, we choose widely used benchmarks in the few-shot named entity (Ding et al., 2021b) and relation extraction (Han et al., 2018; Gao et al., 2019b). For CV, we use classical image classification datasets (Vinyals et al., 2016) in experiments. The results demonstrate that, with only a few additional parameters introduced, such modeling significantly outperforms the baseline. Surprisingly, big prototypes perform extremely well in cross-domain few-shot relation extraction, indicating the promising ability to domain adaptation. Given that such small changes can bring huge benefits, hopefully, big prototypes can inspire new ideas for the research community of representation learning. The source code and checkpoints of our models in experiments will be publicly available for reproducibility.

## 2 RELATED WORK

This work is related to studies of few-shot learning and meta-learning, whose primary goal is to quickly adapt deep neural models to new tasks with few training examples. To achieve that, two branches of studies are proposed: optimization-based methods and metric-based methods. The optimization-based studies (Finn et al., 2017; Franceschi et al., 2018; Ravi & Beatson, 2018) regard few-shot learning as a bi-level optimization process, where a global optimization is conducted to learn a good initialization across various tasks, and a local optimization quickly adapts the initialization parameters to specific tasks with few training examples by few steps of gradient descent.

Compared to the mentioned studies, our work is more related to the metric-based meta-learning approaches (Vinyals et al., 2016; Snell et al., 2017; Satorras & Estrach, 2018; Sung et al., 2018), whose general idea is to learn a metric to measure the similarity between representations and find the closest labeled example (or a derived prototype) for an unlabeled example. Typically, these

methods learn a metric function during episodic optimization. More specifically, we inherit the spirit that uses prototypes to abstractly represent class-level information, which could be tracked back to cognitive science (Reed, 1972; Rosch et al., 1976; Nosofsky, 1986), statistical machine learning (Graf et al., 2009) and similar to the Nearest Mean Classifier (Mensink et al., 2013). In the area of deep learning, Snell et al. (2017) propose the prototypical network to exploit the average of example embeddings as a prototype to perform metric-based classification in few-shot learning. In Prototypical Network, prototypes are estimated by the embeddings of instances, and it is hard to find a satisfying location of the prototypes of the entire dataset. Ren et al. (2018) adapt such prototype-based networks in the semi-supervised scenario where the dataset is partially annotated. A set of prototype-based networks are proposed concentrating on the improvements of prototype estimations and application to various downstream tasks (Allen et al., 2019; Gao et al., 2019a; Li et al., 2019b; Pan et al., 2019; Seth et al., 2019; Ding et al., 2021a; Li et al., 2020c). We discuss big prototypes and some other prototype-enhanced methods in Appendix C.

There has also been a series of works that embed prototypes into a non-Euclidean output space (Mettes et al., 2019; Keller-Ressel, 2020; Atigh et al., 2021). It should be noted that these studies regard hyperspheres or other non-Euclidean manifolds as the embedding space, and our proposed method use hyperspheres to represent big prototypes and conduct metric-based classification in the Euclidean space. Therefore, the focus of our proposed big prototypes is different from the non-Euclidean prototype-based works. We evaluate the effectiveness of big prototypes in three downstream tasks across NLP and CV, including few-shot named entity recognition, few-shot relation extraction, and few-shot image classification. Another technical route to achieve promising results in these downstream few-shot tasks is to use the ability of large-scale pre-trained models (Brown et al., 2020; Han et al., 2021b), or build large-scale data sets to design specific pre-training tasks (Huang et al., 2020; Soares et al., 2019) (which may face the risk of information leakage). These techniques are also orthogonal to our contribution. Generally, the method of big prototypes is model-agnostic.

## 3 BIG PROTOTYPES

This section begins with the problem setup of few-shot learning. Then, we introduce the metrics, initialization, and learning of big prototypes. Generally, unlike previous prototype-based models that use estimated dense vectors as prototypes, our approach use hyperspheres to represent the concept-level information for classes. One big prototype is represented by two parameters: the center and the radius of the hypersphere, which are firstly initialized via estimation and then optimized by gradient descent in an end-to-end fashion.

### 3.1 PROBLEM SETUP

In this work, we consider the episodic $N$ way $K$ shot few-shot learning paradigm. In this setting, we have a large-scale annotated training set $\mathcal{D}_{\text{train}}$, and our goal is to learn a model that could predict for a set of new classes $\mathcal{D}_{\text{test}}$, where only a few examples are labeled.

In this setting, the model will be trained in episodes constructed using $D_{train}$ and tested on episodes constructed using $D_{test}$. Each episode contains a *support* set for learning $\mathcal{S} = \{\mathbf{x}_i, y_i\}_{i=1}^{N \times K}$ with $N$ classes and $K$ examples for each class, and a *query* set for inference $\mathcal{Q} = \{\mathbf{x}_j^*, y_j^*\}_{j=1}^{N \times K'}$ of examples in the same $N$ classes. Each input data is a vector $\mathbf{x}_i \in \mathbb{R}^L$ with the dimension of $L$ and $y_i$ is an index of class label. For each input $\mathbf{x}_i$, let $\mathbf{h}_i = f_\phi(\mathbf{x}_i) \in \mathbb{R}^D$ denote the $D$-dimensional output embedding of a neural network $f_\phi : \mathbb{R}^L \to \mathbb{R}^D$ parameterized by $\phi$.

Beyond the conventional few-shot classification, we also carry out experiments in few-shot named entity recognition. This is a sequence labeling task where each token in a sequence is asked to be labeled as if it is a part of a named entity. But as the context is extremely important for this task, the examples are sampled in sequence-level. Thus the problem setup is slightly different from the traditional $N$ way $K$ shot classification. We follow the strategy in Ding et al. (2021b) and sample sequences in a $N$ way $K \sim 2K$ shot manner (see Appendix B).

### 3.2 METRIC

We now introduce big prototypes, which are a set of hyperspheres in the feature space to abstractly represent the intrinsic features of classes. No matter what the dimension of embedding space is, one

big prototype is represented by $\mathbf{p} = (\mathbf{z}, r)$, where $\mathbf{z} \in \mathbb{R}^D$ is the center of the hypersphere with the same dimension to $\mathbf{h} = f_\phi(\mathbf{x})$, and $r \in \mathbb{R}$ is a scalar denoting the radius of the hypersphere.

The central idea is to learn a big prototype for each class with limited episodic supervision, and each example in the query set $(\mathbf{x}_j^*, y_j^*)$ is predicted by measuring the distance from the embedding $\mathbf{h}^*$ to the surface of the hyperspheres. The distance of two vectors is calculated by a metric function: $d : \mathbb{R}^D \times \mathbb{R}^D \to [0, +\infty)$. In the Euclidean space, the metric being

$$d(\mathbf{h}, \mathbf{z}) = \|\mathbf{h} - \mathbf{z}\|^2. \tag{1}$$

And the distance $\tilde{d}$ from an embedding to a big prototype $\mathbf{p}$ is the distance from the point to the center of the hypersphere minus the radius

$$\tilde{d}(\mathbf{x}, \mathbf{p}) = d(f_\phi(\mathbf{x}), \mathbf{z}) - r = \|\mathbf{h} - \mathbf{z}\|^2 - r. \tag{2}$$

Note that in this case, the value of $\tilde{d}(\cdot)$ may be negative, that is geometrically speaking, the point is contained inside the hypersphere, and it does not affect the calculation of loss. Although more generally, the idea is to use areas but not points in the embedding space to model prototypes, hyperspheres naturally have two obvious advantages. As stated in § 1, one big prototype could be uniquely modeled by the center and the radius $\mathbf{p} = \{\mathbf{z}, r\}$. While characterizing manifolds with complex boundaries in the embedding space is difficult.s Furthermore, it is easy to calculate metrics by Equation 2. We use the distance from a point to the surface as the metric. In this way, the center and the radius are spontaneously in the loss function and optimized. In this geometric interpretation, sparse classes will have larger learned radii, while compact classes will have smaller learned radii.

### 3.3 INITIALIZATION

To construct big prototypes, the first step is the initialization, that is, initialize the center $\mathbf{z}$ and the radius $r$ of the hypersphere. We use subscripts to denote indices of the classes and superscript to denote the indices of the support sets. As stated in § 1, we cannot fathom the distribution of the whole data, thus, we initialize the parameters with the first support set of the $n$-th class $\mathcal{S}_n$

$$\mathbf{z}_n = \frac{1}{|\mathcal{S}_n|} \sum_{(\mathbf{x}_i, y_i) \in \mathcal{S}_n} f_\phi(\mathbf{x}_i), \ r_n = \frac{1}{|\mathcal{S}_n|} \sum_{(\mathbf{x}_i, y_i) \in \mathcal{S}_n} d(f_\phi(\mathbf{x}_i), \mathbf{z}_n), \tag{3}$$

where the center $\mathbf{z}_n$ is initialized by the mean output of the embeddings of the current class like Snell et al. (2017), and $r_n$ is initialized by the average of all the metrics from the embeddings to the center. The initialization essentially approximates the center and the sparse degree of the representations, and is more secure than random initialization. In our experiments, to make it more comparable to previous works, we simply use the first single episode of each class for initialization. However, since the computational cost of this initialization strategy is small, an alternative way is to use the average of the initialized hyperspheres of multiple episodes as the final big prototype. In a $K$-shot episodic training, for the first $m$ support set of the $n$-th class $\mathcal{S}_n^M$, we have

$$\tilde{\mathbf{z}}_n = \frac{1}{m} \sum_{j=1}^m \mathbf{z}_n^j, \ \tilde{r}_n = \frac{1}{m} \sum_{j=1}^m r_n^j, \tag{4}$$

where $m$ is a hyper-parameter. Equation 3 denotes the situation when $m = 1$. And if $m$ is equal to the total number of episodes, establishing big prototypes degenerates to an estimation procedure.

### 3.4 LEARNING

Once initialized, a big prototype will participate in the training process, where its center and radius are simultaneously optimized. A query point $\mathbf{x}$ is classified by softmax over the metrics to the big prototypes calculated by Equation 2

$$p(y = n | \mathbf{x}) = \frac{\exp(\tilde{r}_n - \|\mathbf{h} - \tilde{\mathbf{z}}_n\|_2^2)}{\sum_{n'} \exp(\tilde{r}_{n'} - \|\mathbf{h} - \tilde{\mathbf{z}}_{n'}\|_2^2)}. \tag{5}$$

And the parameters of $f$ and big prototypes are optimized by minimizing the metric-based cross-entropy objective function:

$$\mathcal{L} = -\log p(y | \mathbf{x}, \mathbf{p}). \tag{6}$$

Equation 5 explains the combination of the two advantages of big prototypes, where $\tilde{d}$ is calculated as $r$ and $\mathbf{z}$ will participate in the optimization. The parameters of the neural network are optimized along with the centers and radii of big prototypes through gradient descent. To sum up, in the initialization stage, $\mathbf{z}$ and $r$ are estimated by the embeddings of examples. And in the learning stage, $\mathbf{z}$ and $r$ are optimized by an independent optimizer, because the final location and shape of the hyperspheres need to serve the performance of classification. Algorithm 1 expresses the initialization and learning stages of big prototypes in pseudocode.

The goal of few-shot learning is to learn new classes with a handful of examples. There are two strategies to handle the unseen classes at the episodic evaluation stage. Since there still exist a support set and a query set in evaluation, we could strictly follow the initialization-learning paradigm. Alternatively, we could directly estimate the big prototype like vanilla prototypical network. Keeping aligned with the vanilla prototypes for fair comparison, we use the latter strategy in experiments.

---

**Algorithm 1:** Training process of big prototypes. $f_\phi$ is the feature encoder, $T$ is the total number of samples in the training set, $N_{\text{total}}$ is the total number of classes in the training set, $N$ is the number of classes for support and query set, $K$ is the number of examples per class in the support set, $K'$ is the number of examples per class in the query set, and $M$ is a hyper-parameter. RANDOMSAMPLE$(S, N)$ denotes a set of $N$ elements chosen uniformly at random from set $S$, without replacement.

---

**Input:** Training data $\mathcal{D}_{\text{train}} = \{(\mathbf{x}_1, y_1), ..., (\mathbf{x}_T, y_T)\}$, where each $y_i \in \{1, ..., N_{\text{total}}\}$. $\mathcal{D}_k$
   denotes the subset of $\mathcal{D}$ containing all elements $(\mathbf{x}_i, y_i)$ such that $y_i = k$
**Output:** The updated encoder $f_\phi$
// Initialization phase
**for** $n = 1$ *to* $N_{total}$ **do**
   $\mathcal{S}_n \leftarrow$ RANDOMSAMPLE$(\mathcal{D}_n, K)$
   $\tilde{\mathbf{z}}_n \leftarrow \frac{1}{|\mathcal{S}_n|} \sum\limits_{(\mathbf{x}_i, y_i) \in \mathcal{S}_n} f_\phi(\mathbf{x}_i),$
   $\tilde{r}_n \leftarrow \frac{1}{|\mathcal{S}_n|} \sum\limits_{(\mathbf{x}_i, y_i) \in \mathcal{S}_n} d(f_\phi(\mathbf{x}_i), \mathbf{z}_n),$
// Learning phase
**for** $i = 1$ *to* $M$ **do**
   $V \leftarrow$ RANDOMSAMPLE$(\{1, ..., N_{\text{total}}\}, N), \mathcal{L} \leftarrow 0$
   **for** $n$ *in* $\{1, ..., N\}$ **do**
      $\mathcal{Q}_n \leftarrow$ RANDOMSAMPLE$(\mathcal{D}_{V_n}, K')$
      $\mathcal{L} \leftarrow \mathcal{L} + \frac{1}{NK'} \sum\limits_{(\mathbf{x}_i, y_i) \in \mathcal{Q}_n} [d(f_\phi(\mathbf{x}_i), \mathbf{z}_n) - \tilde{r}_n + \log \sum\limits_{n'} \exp(\tilde{r}_{n'} - d(f_\phi(\mathbf{x}_i), \mathbf{z}_{n'}))]$
   UPDATE $\tilde{\mathbf{z}}, \tilde{r}, f_\phi$ w.r.t $\mathcal{L}$

---

## 4 EXPERIMENTS

To evaluate the effectiveness of the proposed method and compare it with the baselines, we conduct experiments on three common few-shot learning tasks both in NLP and CV, including few-shot named entity recognition (NER), few-shot relation extraction (RE) and few-shot image classification. This section describes and presents the results of each of the three tasks. Note that the goal of big prototypes is to provide a new insight for representation learning of concept-level information, not to lead in all the leaderboards. Since big prototypes are model-agnostic, the central idea of the design for the experiments is to make comparisons with the related baselines. We use original prototypes as our main baseline and give less attention to other orthogonal techniques like the structures of neural models or the task-specific pre-training.

### 4.1 FEW-SHOT NAMED ENTITY RECOGNITION

We assess the effectiveness of big prototypes on NLP, specifically, the first task is few-shot named entity recognition (NER) and the dataset is FEW-NERD (Ding et al., 2021b). Different from typical instance-level classification, few-shot NER is a sequence labeling task, where labels may share structural correlations. NER is the first step in automatic information extraction and the construction of large-scale knowledge graphs. Quickly detecting fine-grained unseen entity types is of significant importance in NLP. To capture the latent correlation, many recent efforts in this field use large pre-

| Setting | Eva. | FEW-NERD (INTRA) | | | FEW-NERD (INTER) | | |
|---|---|---|---|---|---|---|---|
| | | NNShot | Prototypes | Big Prototypes | NNShot | Prototypes | Big Prototypes |
| 5 way 1~2 shot | P | $28.95_{1.02}$ | $18.58_{1.02}$ | $\mathbf{40.18}_{1.71}$ | $50.40_{0.60}$ | $38.70_{0.50}$ | $\mathbf{53.36}_{2.74}$ |
| | R | $\mathbf{33.40}_{1.44}$ | $31.83_{1.03}$ | $26.96_{2.07}$ | $\mathbf{58.84}_{0.13}$ | $52.60_{1.65}$ | $51.12_{4.94}$ |
| | F | $31.01_{1.21}$ | $23.45_{0.92}$ | $\mathbf{32.26}_{1.94}$ | $\mathbf{54.29}_{0.40}$ | $44.58_{0.26}$ | $52.09_{2.49}$ |
| 5 way 5~10 shot | P | $32.87_{2.45}$ | $35.87_{0.69}$ | $\mathbf{48.77}_{0.79}$ | $45.80_{3.53}$ | $53.73_{1.77}$ | $\mathbf{62.26}_{0.89}$ |
| | R | $39.17_{2.17}$ | $50.50_{1.88}$ | $\mathbf{53.26}_{2.60}$ | $56.45_{2.93}$ | $64.99_{2.24}$ | $\mathbf{69.32}_{1.66}$ |
| | F | $35.74_{2.36}$ | $41.93_{0.55}$ | $\mathbf{50.88}_{1.01}$ | $50.56_{3.33}$ | $58.80_{1.42}$ | $\mathbf{65.59}_{0.50}$ |
| 10 way 1~2 shot | P | $20.38_{0.22}$ | $16.52_{0.52}$ | $\mathbf{26.06}_{2.40}$ | $42.74_{2.05}$ | $32.59_{0.22}$ | $\mathbf{45.38}_{0.49}$ |
| | R | $23.63_{0.53}$ | $\mathbf{24.60}_{0.72}$ | $22.32_{0.54}$ | $\mathbf{52.16}_{1.76}$ | $48.91_{2.94}$ | $43.22_{1.33}$ |
| | F | $21.88_{0.23}$ | $19.76_{0.59}$ | $\mathbf{24.02}_{1.06}$ | $\mathbf{46.98}_{1.96}$ | $39.09_{0.87}$ | $44.26_{0.53}$ |
| 10 way 5~10 shot | P | $25.46_{0.63}$ | $28.93_{0.82}$ | $\mathbf{38.94}_{3.39}$ | $45.15_{0.81}$ | $47.93_{0.45}$ | $\mathbf{56.38}_{1.79}$ |
| | R | $30.32_{1.71}$ | $43.08_{0.84}$ | $\mathbf{46.71}_{2.48}$ | $56.05_{0.37}$ | $61.79_{1.73}$ | $\mathbf{65.84}_{1.61}$ |
| | F | $27.67_{1.06}$ | $34.61_{0.59}$ | $\mathbf{42.46}_{3.04}$ | $50.00_{0.36}$ | $53.97_{0.38}$ | $\mathbf{60.73}_{1.47}$ |

Table 1: Performance of state-of-the-art methods on FEW-NERD (INTRA) and FEW-NERD(INTER). In this table, P is precision, R is recall and F refers to F1-score.

trained language models (Han et al., 2021a) like BERT (Devlin et al., 2019) as backbone model and have achieved remarkable performance. The original prototypical network has also been applied to this task (Li et al., 2020b; Huang et al., 2020; de Lichy et al., 2021). We report the dataset, baselines, and experimental details in Appendix A.1.

**Results** Table 1 shows the performance of current state-of-art models on FEW-NERD. Overall, big prototypes have a considerable advantage over prototypes, with an increase of at least 5% in f1-score across all settings. The success on both datasets demonstrates that big prototypes can learn the general distribution pattern of entities across different entity types and thus can greatly improve the performance when little information is shared between training and test set. It can also be observed that a large portion of the improvement comes from the increase in precision, showing that big prototypes can better distinguish entities from context. It is possibly because context words are much diverse and modeling them with a hypersphere as in big prototypes is more plausible than a single point as in prototypes. With respect to number of shots, big prototypes are more advantageous when larger shots are provided and becomes the new state-of-art in the 5~10 shot setting. For the comparison with NNShot, big prototypes remains the superiority under the settings of high-shot (5~10), outperforming it by at least 10% of F1-scores. Interestingly, the performances of NNShot and big prototypes are comparable when it comes to low-shot. This is because in the sequece labeling task, it is more difficult to infer the class-level information from very limited tokens. And the use of big prototypes and tokens for metric-based classification play a similar role. And the mediocre performance on low shot settings indicates that the hypersphere parameters are learned more effectively when more shots are available. Also since in strict 1-shot setting the initialization of the radius would be zero, we believe a good initialization strategy is important for big prototypes.

## 4.2 FEW-SHOT RELATION EXTRACTION

The other common NLP task is relation extraction (RE), which aims at correctly classifying the relation between two given entities in a sentence. This is a fundamental task in information extraction. RE is an important form of learning structured knowledge from unstructured text. We use FewRel Han et al. (2018) and FewRel 2.0 Gao et al. (2019b) as the datasets. In real-world datasets, many of the relations are long-tailed and thus cannot be identified accurately under the common supervised setting. Traditional methods often alleviate the problem with distant supervision, which would result in wrong labels. Recent approaches have applied few-shot learning models on the task to learn from a handful of samples, which yield promising results (Gao et al., 2019a). We report the datasets, baselines and experimental details in Appendix A.2.

**Results** Table 2 presents the results on two FewRel tasks, we did not compare methods that use additional data or conduct task-specific encoder pre-training. Big prototypes generally perform better than prototypes across all settings. In terms of backbone models, when combining with pre-trained models like BERT, big prototypes can yield a larger advantage against prototypes. It shows

| Model | FewRel 1.0 | | | |
|---|---|---|---|---|
| | 5 way 1 shot | 5 way 5 shot | 10 way 1 shot | 10 way 5 shot |
| Meta Net (Munkhdalai & Yu, 2017) | 64.46 ± 0.54 | 80.57 ± 0.48 | 53.96 ± 0.56 | 69.23 ± 0.52 |
| SNAIL (Mishra et al., 2017) | 67.29 ± 0.26 | 79.40 ± 0.22 | 53.28 ± 0.27 | 68.33 ± 0.26 |
| GNN CNN (Satorras & Estrach, 2018) | 66.23 ± 0.75 | 81.28 ± 0.62 | 46.27 ± 0.80 | 64.02 ± 0.77 |
| GNN BERT (Satorras & Estrach, 2018) | 75.66 ± 0.00 | 89.06 ± 0.00 | 70.08 ± 0.00 | 76.93± 0.00 |
| Proto-HATT (Gao et al., 2019a) | 76.30 ± 0.06 | 90.12 ± 0.04 | 64.13 ± 0.03 | 83.05 ± 0.05 |
| MLMAN (Ye & Ling, 2019) | 82.98 ± 0.20 | 92.66 ± 0.09 | 73.59 ± 0.26 | 87.29 ± 0.15 |
| Proto CNN | 69.20 ± 0.20 | 84.79 ± 0.16 | 56.44 ± 0.22 | 75.55 ± 0.19 |
| Big Proto CNN (Ours) | 66.05 ± 3.44 | 87.31 ± 0.93 | 56.74 ± 1.06 | 77.87 ± 2.60 |
| Proto BERT | 80.68 ± 0.28 | 89.60 ± 0.09 | 71.48 ± 0.15 | 82.89 ± 0.11 |
| Big Proto BERT (Ours) | **84.34 ± 1.23** | **93.42 ± 0.50** | **77.24 ± 6.05** | **88.71 ± 0.31** |
| | FewRel 2.0 Domain Adaptation | | | |
| Proto-ADV CNN (Wang et al., 2018) | 42.21 ± 0.09 | 58.71 ± 0.06 | 28.91 ± 0.10 | 44.35 ± 0.09 |
| Proto-ADV BERT (Gao et al., 2019b) | 41.90 ± 0.44 | 54.74 ± 0.22 | 27.36 ± 0.50 | 37.40 ± 0.36 |
| BERT-pair (Gao et al., 2019b) | 56.25 ± 0.40 | 67.44 ± 0.54 | 43.64 ± 0.46 | 53.17 ± 0.09 |
| Proto CNN | 35.09 ± 0.10 | 49.37 ± 0.10 | 22.98 ± 0.05 | 35.22 ± 0.06 |
| Big Proto CNN (Ours) | 36.41 ± 1.43 | 55.50 ± 1.42 | 22.11 ± 0.58 | 40.82 ± 2.50 |
| Proto BERT | 40.12 ± 0.19 | 51.50 ± 0.29 | 26.45 ± 0.10 | 36.93 ± 0.01 |
| Big Proto BERT (Ours) | **59.03 ± 3.68** | **74.85 ± 4.59** | **45.88 ± 7.43** | **61.61 ± 4.69** |

Table 2: Accuracies of Prototypes and Big Prototypes on FewRel 1.0 and FewRel 2.0 under 4 different settings. Separate blocks show the direct comparisons between prototypes and big prototypes.

that the distribution assumption of big prototypes may be closer to the real data distribution, which boosts the finetuning phase. Meanwhile it also sheds light on the untapped potential ability of large pre-trained models and stresses that the assumptions made about data distribution may help us unlock the potential. It is also worth noting that big prototypes achieves outstanding performance on Domain Adaptation task, with an increase of about 20% in accuracies when using BERT. The success signals the great capability of big prototypes in transfer learning and also points to the importance of a better abstraction of data distribution in transfer learning investigation. Although big prototypes yield surprisingly performance in the domain adaptation task, the standard errors are also high. This observation suggests that when the domain shifts, the unstable encoding of examples by language model may lead to different results. Compared to other meta learning baselines, big prototypes also yield promising results on all the settings.

### 4.3 FEW-SHOT IMAGE CLASSIFICATION

Image classification is one of the most classical tasks in few-shot learning research. Seeking a better solution for few-shot image classification is beneficial in two ways: (1) to alleviate the need for data augmentation, which is a standard technique to enrich the labeled data by performing transformations on a given image; (2) to facilitate the application where the labeled image is expensive. We use *mini*ImageNet (Vinyals et al., 2016) as the dataset in our experiment. The dataset, baselines, and experimental details are reported in Appendix A.3.

**Results** Table 3 shows the result on *mini*ImageNet few-shot classification under 2 settings. Big prototypes substantially outperform prototypes in most settings, displaying their ability in modeling class distribution of images. We observe that compared to NLP, the results of image classification are more stable both for vanilla prototypes and big prototypes. This observation may indicate the difference in encoding between the two technologies. Token representations in BERT are contextualized and changeable around different contexts, yet the image representation produced by deep CNNs aims to thoroughly capture the global and local features. Different from the NLP experiments, we use 5-way 1-shot and 5-way-5-shot settings in image classification. Under the 5-way 5-shot setting, the improvements of big prototypes are significant. Compared to the main baseline of prototypes, big prototypes could yield considerable improvements with all the three backbones. The effectiveness of our method is also demonstrated by the comparisons with other previous few-shot learning methods. With different backbones, big prototypes achieve better or comparable results than the pre-

vious state-of-the-art models. In particular, with the WideResNet (Zagoruyko & Komodakis, 2016) backbone, big prototypes yield the best results of all the compared methods, suggesting that as the representational power of the model improves, so does the expressive capability of big prototypes. Compared to the 5-shot setting, the big prototypes achieve less boost in the 1-shot setting of Convnet and ResNet-12 (He et al., 2015), the phenomenon is consistent with the intuition that more examples would be more favourable to the learning of radius.

| Model | Backbone | *mini*ImageNet | |
|---|---|---|---|
| | | **5 way 1 shot** | **5 way 5 shot** |
| IMP (Allen et al., 2019) | ConvNet | 33.30 ± 0.71 | 65.88 ± 0.71 |
| Prototypes (Snell et al., 2017) | ConvNet | 46.44 ± 0.60 | 63.72 ± 0.55 |
| CovaMNet (Li et al., 2019a) | ConvNet | 51.83 ± 0.64 | 65.65 ± 0.77 |
| Big Prototypes (Ours) | ConvNet | 50.21 ± 0.31 | 66.48 ± 0.71 |
| SNAIL (Mishra et al., 2017) | ResNet-12 | 55.71 ± 0.99 | 68.88 ± 0.92 |
| Prototypes (Snell et al., 2017) | ResNet-12 | 53.81 ± 0.23 | 75.68 ± 0.17 |
| Variational FSL (Zhang et al., 2019) | ResNet-12 | 61.23 ± 0.26 | 77.69 ± 0.17 |
| Prototypes + TRAML (Li et al., 2020a) | ResNet-12 | 60.31 ± 0.48 | 77.94 ± 0.57 |
| Meta-baseline (Chen et al., 2021) | ResNet-12 | 63.17 ± 0.23 | 79.26 ± 0.17 |
| Big Prototypes (Ours) | ResNet-12 | 59.65 ± 0.62 | 78.24 ± 0.47 |
| Prototypes (Snell et al., 2017) | WideResNet-28-10 | 59.09 ± 0.64 | 79.09 ± 0.46 |
| Activation to Parameter (Qiao et al., 2018) | WideResNet-28-10 | 59.60 ± 0.41 | 73.74 ± 0.19 |
| LEO (Rusu et al., 2018) | WideResNet-28-10 | 61.76 ± 0.08 | 77.59 ± 0.12 |
| Big Prototypes (Ours) | WideResNet-28-10 | **63.78 ± 0.63** | **81.29 ± 0.46** |

Table 3: Accuracies with 95% confidence interval of big prototypes and baselines on *mini*ImageNet
.

## 4.4 ANALYSIS

**Radius Dynamics** In this subsection, the mechanism of big prototypes will be empirically analyzed. We demonstrate the mechanism of big prototypes by illustrating the change of radius for one specific hypersphere. In the learning phase, the radius of a big prototype is changing according to the "density" of the sampled episode, which could be quantified by the mean distance of samples to the corresponding prototype center. Practically, since the sampling is random, the value of the mean distance may oscillate at a high frequency in this process, and the radius changes accordingly.

To better visualize the changing of radius along with the mean distance at each update, for each round of training we fix one specific class as the *anchor class* for mean distance and radius recording and apply a special sampling strategy at each episode. Specifically, we take FewRel training data and train on the 5 way 5 shot setting with CNN encoder. While training, each episode contains the *anchor class* and 4 other randomly sampled classes. Training accuracy is logged every 50 steps. After a warmup training of 500 steps, we sample "good" or "bad" episodes for every 50 steps alternatively. A "good" episode has higher accuracy on the anchor class than the previously logged accuracy, while conversely, a "bad" episode has an accuracy lower than before. The mean distance to the prototype center and radius at each episode are logged every 50 steps after the warmup.

Figure 1 shows the changing of mean distance and radius for 8 classes during 600∼2000 training steps. Although the numeric values of distance and radius differ greatly and oscillate at different scales, they have similar changing patterns. Besides, it could be observed that there is often a small time lag in the change of radius, indicating that the change of radius is brought by the change in mean distance. This is in line with our expectations and perfectly demonstrates the learning mechanism of big prototypes. The experiment provides a promising idea, if we can control the sampling strategy through priori knowledge, we may find a way to learn ideal big prototypes.

**Visualization** We also use $t$-SNE (van der Maaten & Hinton, 2008) to visualize the embedding before and after training. 5 classes are sampled from training set and test set of Few-NERD dataset, respectively, and for each class 500 samples are randomly chosen to be embedded by BERT trained on 5 way 5 shot NER task. Figure 2 is a graph showing the result of embeddings in a 2-dimensional space, where each distinct color represents one class. Note that for NER task the representations are obtained on the token level, where the interaction between the target token and its context may

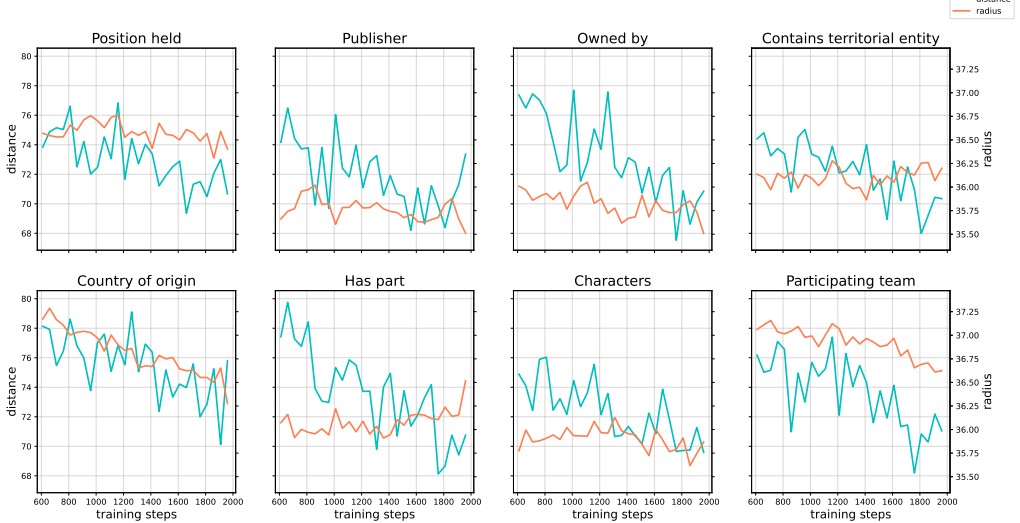

Figure 2: The illustration depicts the radius change according to the degree of sparsity of the sampled episode. Each subfigure represents a selected anchor class in the FewRel dataset.

result in a more mixed-up distribution compared to instance-level embedding. It could be seen that after training, the representations of the same class in the training set become more compressed and easier to classify. It is also worth noting that the trained BERT encoder could even generate a more clustered representation distribution for a novel class. The visualization demonstrates the effectiveness of big prototypes training in learning discriminative features, especially the success on learning novel class representation that greatly boosts model performance under few-shot settings.

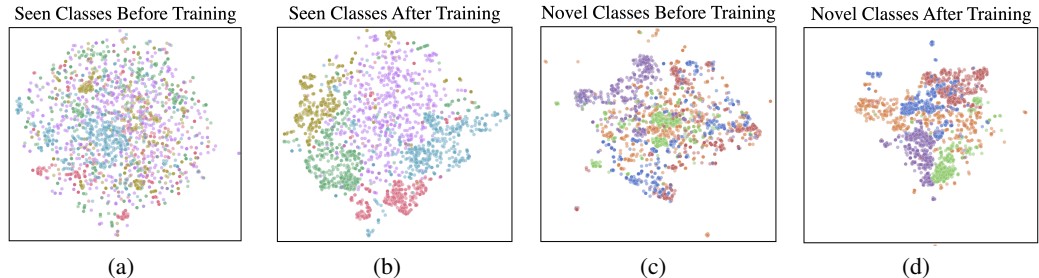

Figure 3: $t$-sne visualization of feature distributions. (a) illustrates the representations of seen data (in training data) before training; (b) illustrates the representations of seen data (in training data) produced by our network; (c) illustrates novel data (in test data) before training; (d) illustrates novel data (in test data) produced by our network after training. Note that even after training, the neural network has never seen the novel data and their classes.

## 5 CONCLUSION

This paper proposes a novel metric-based few-shot learning method, *big prototypes*. Unlike previous prototype-based methods that use dense vectors to represent the class-level semantics, we use hyperspheres to enhance the capabilities of prototypes to express the intrinsic information of the data. It is simple to model a hypersphere in the embedding space than other complex manifolds, we establish two variables, the center and the radius to represent a big prototype. Such modeling is easy to implement and also empirically effective, we observe significant improvements than vanilla prototype on three tasks across NLP and CV. For potential future work, big prototypes could be extended to more generalized representation learning like image, word, or sentence representations.

## 6 ETHICS STATEMENT

This section states the ethical considerations of our work. Generally, the method of big prototypes is a tool for few-shot learning, making it intrinsically unbiased. The impact, no matter positive or negative, is depend on the intentions of specific applications that uses our method. Nevertheless, since few-shot systems have already been deployed in real-world scenarios, it is crucial to carefully choose the training data to avoid the potential bias. We strongly encourage the users of our method to be careful about the data selection in practice. All the data used in our research are granted for research purpose, and it will not result in privacy issues. In terms of energy saving, we will make all the checkpoints in our experiments publicly available to reduce the carbon emission.

## 7 REPRODUCIBILITY STATEMENT

We evaluate the effectiveness of big prototypes in three tasks across NLP and CV, which are few-shot named recognition, few-hot relation extraction, and few-shot image classification (§ 4). For each experiment, we provide an experimental setting part to introduce details for reproduction, including data processing, hyper-parameters, the choice of the neural encoder, and the steps for training, validating, and testing. Please refer to § 4.1, § 4.2, § 4.3 for the details of the three tasks, respectively. For the specific initialization and learning of big prototypes, we provide pseudocode in Algorithm 1. The source code and the checkpoints in our work will be released.

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

## A EXPERIMENTAL DETAILS

This section reports the experimental details of all the three tasks in our evaluation. All the experiments are conducted on NVIDIA A100 and V100 GPUs with CUDA.

### A.1 EXPERIMENTAL DETAILS FOR FEW-SHOT NAMED ENTITY RECOGNITION

**Dataset** The experiment is run on FEW-NERD dataset (Ding et al., 2021b). It is a large-scale NER dataset containing over 400,000 entity mentions, across 8 coarse-grained types and 66 fine-grained types, which makes it an ideal dataset for few-shot learning. It has been shown that existing methods including prototypes are not effective enough on this dataset.

**Baselines** NNShot (Yang & Katiyar, 2020) is a token-level metric-based method that is specifically designed for few-shot labeling. Note that the main baseline here is the Proto method, which adapts the prototypical network on few-shot named entity recognition.

**Implementation Details** We run experiments under four settings on the two released benchmarks, FEW-NERD (INTRA) and FEW-NERD (INTER). Specifically, we use uncased BERT as the backbone encoder and 1e-4 as the encoder learning rate. We manually tune the learning rate for the radius parameter and the best result is obtained with 10. AdamW is used as the BERT optimizer and Adam (Kingma & Ba, 2017) is used to optimize prototype radius. The batch size is set to 2 across all settings. All models are trained for 10000 steps and validated every 1000 steps. The results are reported on 5000 steps of the test episode. For each setting, we run the experiment with 3 different random seeds and report the average metrics including precision, recall, f1-score, and the standard error for each. We use PyTorch (Paszke et al., 2019) and huggingface transformers (Wolf et al., 2020) to implement the backbone encoder $BERT_{base}$.

### A.2 EXPERIMENTAL DETAILS FOR FEW-SHOT RELATION EXTRACTION

**Dataset** We adopt the FewRel dataset (Han et al., 2018; Gao et al., 2019b), a relation extraction dataset specifically designed for few-shot learning. FewRel has 100 relations with 700 labeled instances each. The sentences are extracted from Wikipedia and the relational entities are obtained from Wikidata. FewRel 1.0 (Han et al., 2018) is released as a standard few-shot learning benchmark. FewRel 2.0 (Gao et al., 2019b) adds domain adaptation task and NOTA task on top of FewRel 1.0 with the newly released test dataset on PubMed corpus.

**Baselines** In addition to the main baseline, prototypical network Snell et al. (2017), we also choose the following few-shot learning methods as the baselines in relation extraction. (1) Proto-HATT Gao et al. (2019a) is a neural model with hybrid prototypical attention. (2) MLMAN Ye & Ling (2019) is a multi-level matching and aggregation network for few-shot relation classification. Note that Proto-HATT and MLMAN are not model-agnostic. (3) GNN Satorras & Estrach (2018) is a meta-learning model with a graph neural network as the prediction head. (4) SNAIL Mishra et al. (2017) is a meta-learning model with attention mechanisms. (5) Meta Net Munkhdalai & Yu (2017) is a classical meta-learning model with meta information. (6) Proto-ADV Gao et al. (2019b) is a prototype-based method enhanced by adversarial learning. (7) BERT-pair Gao et al. (2019b) is a strong baseline that uses BERT for few-shot relation classification. We re-run all the baselines, except for MLMAN and report the corresponding performances.

**Implementation Details** The experiments are conducted on FewRel 1.0 and FewRel 2.0 domain adaptation tasks. For FewRel 1.0, we follow the official splits in Han et al. (2018). For FewRel2.0, we follow Gao et al. (2019b), training the model on wiki data, validating on SemEval data, and testing on the PubMed data. We use the same CNN structure and BERT as encoders. The learning rate for big prototype radius is 0.1 and 0.01 for CNN and BERT encoder, respectively. Adam (Kingma & Ba, 2017) is used as radius optimizer. We train the model for 10000 steps, validate every 1000 steps, and test for 5000 steps. The other hyperparameters are the same as the original paper.

### A.3 EXPERIMENTAL DETAILS FOR FEW-SHOT IMAGE CLASSIFICATION

**Dataset** *mini*ImageNet (Vinyals et al., 2016) is used as a common benchmark for few-shot learning. The dataset is extracted from the full ImageNet dataset (Deng et al., 2009), and consists of 100

randomly chosen classes, with 600 instances each. Each image is of size $3\times84\times84$. We follow the split in (Ravi & Larochelle, 2017) and use 64, 16, and 20 classes for training, validation, and testing.

**Baselines** The baselines we choose are as follows: (1) Prototypical network (Snell et al., 2017) is our main baseline; (2) IMP (Allen et al., 2019) is a prototype-enhanced method that models infinite mixture of prototypes for few-shot learning; (3) CovaMNet (Li et al., 2019a) is a few-shot learning method that uses covariance to model the distribution information to enhance few-shot learning peformance. (4) SNAIL (Mishra et al., 2017) is an attention-based classical meta-learning method; (5) Variational FSL (Zhang et al., 2019) is a variational Beyasian framework for few-shot learning, which contains a pre-training stage; (6) Activation to Parameter (Qiao et al., 2018) predicts parameters from activations in few-shot learning; (7) LEO (Rusu et al., 2018) optimizes latent embeddings for few-shot learning. (8) TRAML (Li et al., 2020a) uses adaptive margin loss to boost few-shot learning, and Prototypes + TRAML is a strong baseline in recent years.; (9) Meta-baseline Chen et al. (2021) is a pre-training & tuning method that serves as a strong baseline in few-shot learning.

**Implementation Details** The experiments are conducted on 5 way 1 shot and 5 way 5 shot settings. To ensure validity and fairness, we implement big prototypes with various backbone models including CNN, ResNet-12 and WideResNet (Zagoruyko & Komodakis, 2016) to make it comparable to all baseline results, and we also re-run some of the baselines including prototypical network (Snell et al., 2017), infinite mixture prototypes (Allen et al., 2019), and CovaMNet (Li et al., 2019a) under our settings based on their original code. Other baseline results are taken from the original paper. Each model is trained on 10,000 randomly sampled episodes for 30~40 epochs and tested on 1000 episodes. The result is reported with 95% confidence interval. Note that both ResNet-12 and WideResNet (Zagoruyko & Komodakis, 2016) are pretrained on the training data, where the pretrained ResNet-12 is identical to Chen et al. (2021) and the pretrained WideResNet follows Mangla et al. (2020). The CNN structure is the same as Snell et al. (2017), which is composed of 4 convolutional blocks each with a 64-filter $3 \times 3$ convolution, a batch normalization layer (Ioffe & Szegedy, 2015), a ReLU nonlinearity, and a $2 \times 2$ max-pooling layer. We use SGD optimizer for the encoder and Adam (Kingma & Ba, 2017) optimizer for the prototype radius. The learning rate for the backbone model is 1e-3. The learning rate for radius is manually tuned and the reported result has a learning rate of 10. At the training stage, the prototype center is re-initialized at each episode as the mean vector of the support embeddings.

## B K~2K Sampling for Few-NERD

In the sequence labeling task FEW-NERD, the sampling strategy is slightly different from other classification task. We follow the strategy of the original paper Ding et al. (2021b) and report it in Algorithm 2.

---

**Algorithm 2:** Greedy $N$-way $K\sim2K$-shot sampling algorithm for FEW-NERD

---

**Input:** Dataset $\mathcal{X}$, Label set $\mathcal{Y}$, $N$, $K$
**Output:** output result
$\mathcal{S} \leftarrow \varnothing$; // Init the support set
// Init the count of entity types
**for** $i = 1$ *to* $N$ **do**
  $\quad$ Count$[i] = 0$ ;
**repeat**
  $\quad$ Randomly sample $(\boldsymbol{x}, \boldsymbol{y}) \in \mathcal{X}$ ;
  $\quad$ Compute $|$Count$|$ and Count$_i$ after update ;
  $\quad$ **if** $|Count| > N$ *or* $\exists Count[i] > 2K$ **then**
  $\quad\quad$ Continue ;
  $\quad$ **else**
  $\quad\quad$ $\mathcal{S} = \mathcal{S} \bigcup (\boldsymbol{x}, \boldsymbol{y})$ ;
  $\quad\quad$ Update Count$_i$ ;
**until** $Count_i \geq K$ *for* $i = 1$ *to* $N$;

---

## C OTHER PROTOTYPE-ENHANCED METHODS

Here, we discuss the difference between big prototypes with four prototype-enhanced methods in few-shot learning: infinite mixture prototypes (Allen et al., 2019), CovaMNet (Li et al., 2019a), Variational Few-Shot Learning (Zhang et al., 2019), and Two-Stage Approach (Das & Lee, 2020).

Infinite mixture prototypes (Allen et al., 2019) model each class as an indefinite number of clusters, and the prediction is obtained by computing and comparing the distance to the nearest cluster in each class. This method is an intermediate model between prototypes and nearest neighbor model, whereas big prototypes alleviate the over-generalization problem of vanilla prototypes with a single additional parameter that turns a single point modeling into a hypersphere. The essential prototype-based feature of big prototypes allows faster computation and easier training.

CovaMNet (Li et al., 2019a) calculates local variance for each class based on support samples and conduct metric-based learning via covariance metric, which basically evaluates the cosine similarity between query samples and the eigenvectors of the local covariance matrix. To ensure the non-singularity of covariance matrix, the feature of each sample is represented with a matrix, generated by a number of local descripters, with each extracting a feature vector. Compared to big prototypes, both methods attempt to model more variance based on vanilla prototypes, while the idea of big prototypes is more straightforward and requires fewer parameters. On the other hand, the multi-channel features adopted by CovaMNet are less natural for NLP tasks.

Variational Few-Shot Learning (Zhang et al., 2019) tackles few-shot learning problem under a bayesian framework. In order to improve single point based estimation, a class-specific latent variable representing the class distribution is introduced and is assumed to be Gaussian. The method parameterizes the mean and variance of the latent variable distribution with neural networks that take the feature of a single instance as input. The learning and inference process are both conducted on the latent variable level. The method adopts variational inference and is built on modeling distribution as a latent variable, where the metric calculation highly relies on the Gaussian assumption. Big prototypes, on the other hand, models the distribution with a center vector and a radius parameter in the actual embedding space, which is more tangible and easier to calculate.

Two-Stage Approach first trains feature encoder and variance estimator on training data in episodic manner with extracted absolute and relative features. Then in the second stage, training data are split into "novel" class and base class, novel class prototypes are learned from both sample mean and base class features. The classification is carried out with the integrated prototypes. This method improves on vanilla prototypes by extracting more features and combining information from base class, but still follows single-point-based metric learning. Big prototypes extends single point to a hypersphere in the embedding space, and therefore better captures within-class variance.

