# OpenReview forum: "Few-shot Learning with Big Prototypes"
_ICLR.cc/2022/Conference — ICLR 2022 Submitted_

### Official Review · Reviewer_7SK7 · 2021-10-30

**Correctness:** 3
**Technical Novelty And Significance:** 2
**Empirical Novelty And Significance:** 2
**Recommendation:** 6
**Confidence:** 4

**Main Review:**

# Strength:
- evaluation on a variety of tasks from different domains
- simple, intuitive, well-justified idea

# Weakness:
- slim contribution:  addition of one parameter to an existing network
- only tested out on a single prototype network from 2017, which hasn't been SOTA for a long time. No comparisons to more recent prototype baselines [1,2] or approaches such as meta-learning [3], or self-supervision [4]. This lack of comparison makes the claims much less substantial. Big prototypes improve on vanilla prototypes, but do they make Euclidean prototypes competitive wrt modern methods?
- While the writing is generally clear, there are many mistakes and at times clumsy formulations getting in the way of clarity and rigor
- The way the prototypes are initialized (3.3) is not very clear to me at all. What is the benefit of this approach instead of simply computing the average center and radius? And if it is a contribution, why isn't it evaluated?


# Suggestions:
The authors could turn the extreme simplicity of their approach into a strength if they adapted their method to a variety of prototype-based algorithms, including more recent and better-performing ones. Spheres exist in hyperbolic space, for example.

# Details:

**abstract:**
- dense vectors = prototype?
- the radius of the sphere is not constant if I understand it properly
- what is an atactic manifold?
- the authors are constantly the world "metric" to mean "distance in a metric space." This makes for some confusing sentences
introduction:
- what is the meaning of derivative in  "derivative prototype-based methods"?
- how are Big prototypes easier to "model"?

**Fig1:**
- the fact that the embeddings change as well makes the figures hard to understand.
- are the center and radius really learned separately and not end-to-end? This is not my understanding from the rest of the paper
Related work
- type of NMS
-  "find the true location of the prototypes of the entire dataset" I don't think there is such a thing as a true embedding for a prototype.

**3 Methods**
-  "One big prototype is represented by two terms" -> parameters
- why limit yourself to the case in which the data is a vector of dimension L? it could be an image, for example
- "We now introduce big prototypes, which are a series of hyperspheres " -> a set of hyperspheres
- 3.3 is quite confusing to read. For example, z^j_n is never defined.

Equation 5: \tilde{r} is indexed by n, and the sum at the denominator is not indexed. It makes it look like the \tilde{r} could be canceled out, when it is obviously not the case

**Algo1:**

Init: And why not just average over Dn?

**Table1**
 P,R,F never defined. "spaticularly"

[1] Gao et al.  Hybrid attention-based prototypical networks for noisy few-shot relation classification. AAAI, 2019.

[2] Mettes et al. Hyperspherical prototype networks. NeurIPS 2019

[3] Rusu et al.  Meta-learning with latent embedding optimization. ICLR 2019

[4] Gidaris et al. Dynamic few-shot visual learning without forgetting CVPR 2018





**Summary Of The Paper:**

The authors propose to augment an existing prototype-based few-shot learning approach by augmenting each prototype with a radius. They detail varied numerical experiments showing the advantage provided by this addition.

**Summary Of The Review:**

A good idea with an interesting multi-domain evaluation. However, the lack of comparison with modern approaches and the use of a single backbone from 4 years ago makes the method not as convincing as it could be. Ultimately, this is not enough for ICLR.

I strongly encourage the authors to adapt their simple idea to a wider range of methods (including hyperbolic prototypes) and add more baselines (and also significantly polish the writing and rigor). There is a potentially very good paper to come from this.

Post discussion: the authors have addressed my concerns and have added many experiments to better substantiate their claim. In the end, a simple yet interesting idea.

---

> ### Author Response · Authors · 2021-11-17
> **Response to Reviewer 7SK7 (2/2)**
>
> ### Only Addition of One Parameter
>
> - We respectfully think simplicity is an advantage but not a weakness of big prototypes. The fact that a single parameter can make a considerable difference is promising in itself.
>
>
> ### Writing & Details
>
> - We have modified the paper in the revised version according to your suggestions, thanks.
> - About referring to prototypes as dense vectors. Prototypes are often formulated as a single vector learned from support samples in few-shot settings. In the vanilla prototypes method, it is the mean of support sample embeddings, which is a dense vector in the embedding space. We emphasize this feature to distinguish our big prototypes which involve more than one single vector from the original method.
> - The expression "derivative prototype-based methods" generally refers to the group of methods that inherit the idea of prototypes.
>
>
> ### Adapt to more scenarios
>
> - Spheres existing in hyperbolic space is an interesting idea, and this suggestion is kind of similar to the comment of Reviewer 6kXL who is interested in the cosine metric. In the response to Reviewer 6kXL, we construct cone-like big prototypes and use the cosine similarity as the metric function. We will empirically verify the approach.

---

> > ### Comment · Reviewer_7SK7 · 2021-11-19
> > **Questions remaining about the initialization strategy**
> >
> > I want to thank the authors for the impressive work in their response. The paper is much more convincing now.
> >
> > I still have reservations about the Initialization. If I follow correctly, the authors propose to initialize the prototype from m random sampling in a row and before the start of the episode. This raises several concerns for me:
> > - this does not seem in the spirit of few-shot learning. Depending on the choice of m (which I could not find in the implementation details), the support set may end up being all the samples in the dataset! This would be problematic.
> > - following  Algorithm 1, it seems that the query set and the support set are not disjoint in general. Is this on purpose? This would go against the principle of episodes.
> > - the authors claim that their Initialization is a "more secure [?] initialization strategy than random initialization" but never assess its quantitative impact. I would be interested in comparing with the classic setting in which the support set and the query set are distinct.
> >
> > The authors still have not changed their Equation 5 in which the subscripts n and n' are missing from \tilde{r}, making it look like \tilde{r} could be canceled out.
> >
> > Should the author answer/correct my last concern, I am willing to improve my rating to an acceptance.

---

> > > ### Author Response · Authors · 2021-11-19
> > > **Further response to Reviewer 7SK7**
> > >
> > >
> > >
> > > Thanks for the comment! Sorry about the confusion made in experimental settings. We will clarify it here.
> > >
> > > ### Few-shot Setting
> > > - We follow the classical N-way-K-shot settings as all the previous works do. To begin with, there are two datasets with **disjoint label space**, a large training dataset $D_{train}$ and a test dataset $D_{test}$. The aim of few-shot learning is to train a model on $D_{train}$ that also performs well on $D_{test}$
> > > - When training on $D_{train}$, we follow the episodic way as previous works do. Each episode consists of one support set $S=\{x_i, y_i\}^{N\times K}$ and a query set $Q=\{x_i, y_i\}^{N \times K}$. The support set contains $N$ classes, each with $K$ instances, and so does the query set. Note that support set and query set have the same label space. In each episode, the model learns on the few samples provided in the support set and inferences on the query set.
> > > - When the training is finished, we test the model in the same episodic manner, and the result is reported on the query set in test episodes (i.e. the query set of ${D_{test}$). It is also in line with the previous works.
> > > - To sum up, the training data $D_{train}$ and the test data $D_{test}$ have disjoint labels, and for an episode (in $D_{train}$ or $D_{test}$), the support set and the query set will have same label space.
> > >
> > >
> > > ### About the initialization strategy and hyper-parameter $m$
> > > - In vanilla prototypes, the "training" on support set is basically calculating the mean vector for each class in the support set based on instance representation.
> > > - In big prototypes, we do the same thing in the test phase. The prototype center and radius are calculated as the mean vector of representation and the mean distance to the center for each class.
> > > - In the training phase of big prototypes, with a large amount of data available, we instead formulate it as a continuous optimization problem, i.e. the prototype center and radius for each class are firstly initialized by support set in the first $m$ episodes and later being continuously updated with loss calculated on query set in each episode. We believe this initialization strategy is more reasonable than random initialization. Algorithm 1 illustrates the training process.
> > > - Since it is only used for training and $D_{train}$ and $D_{test}$ do not share common labels, the setting of $m$ will not violate the N-way-K-shot setting in test phase. In our experiment, we set $m=1$. Thanks for your comment, we have added this information to the experimental details.
> > > - To clarify, we do not use any additional supervision in the training phase, and in the testing phase it remains exactly the same as the previous FSL methods.
> > >
> > > ### Equation 5
> > >
> > > - Thanks for pointing out the omission of subscripts in equation (5), we have fixed that.

---

> > > > ### Comment · Reviewer_7SK7 · 2021-11-20
> > > > **Follow up on initialization**
> > > >
> > > > Thank you I think I understand better, and you have reassured me regarding the few shot setting. If I followed correctly, m corresponds to a set of episodes in which the prototypes are defined directly from the moments of the distribution. Then M is the number of gradient-based episodes. Presenting it as such may make these hyper parameters more intuitive.
> > > >
> > > > One remarks still: if you only use m=1 in all your experiments (which boils down to classic moment-based initialization), I find it disingenuous to present your initialization strategy in detail as a contribution that is more  "secure" and "reasonable" than the alternative. Either actually assess quantitatively the benefit of your strategy (m=0: random, m=1: classic, m>1: yours) or remove it altogether.

---

> > > ### Author Response · Authors · 2021-11-20
> > > **Further response to Reviewer 7SK7 on initialization**
> > >
> > > Thanks for the comment, now we have uploaded a new revision.
> > > - First, we would like to clarify that we did not intend to present the details of initialization as a contribution. The reason we introduce initialization is that we want to present a full cycle of the parameters of big prototypes. We also cited [Snell et al.2017] of the initialization of the center to indicate that it is not our core contribution. The contributions of this paper are (1) the modeling of big prototypes itself, and (2) the cross-task evaluation.
> > > - In the paper, "we choose a more secure initialization strategy than random initialization" means this average initialization strategy is intuitively more secure than **random initialization** and does not intend to emphasize other strategies.
> > > - Sorry for the confusion, we introduce $m$ only to show another more general way to initialize the parameters. We agree that we use classic moment-based initialization ($m=1$) in our experiments. And to further clarify our setting, we remove $m$ in Algorithm 1 and further elucidate that we only use the first episode of each class for initialization. Thanks again!

---

> > > > ### Comment · Reviewer_7SK7 · 2021-11-21
> > > > **Acknowledgement**
> > > >
> > > > Thank you for your response. My concerns have all been addressed, and I think the paper can now be published. The idea is simple, but it does seem to help.

---

> ### Author Response · Authors · 2021-11-17
> **Response to Reviewer 7SK7 (1/2)**
>
> Sincerely thanks for the valuable comments and suggestions.
>
> ### More baselines
>
> - We agree that more baselines need to be compared and big prototypes could be adapted to more scenarios. Here are the empirical results.
> - For image classification, we add the following baselines, IMP (infinite mixture of prototypes) [1], CovaMNet [2], Variational FSL [3], LEO[5],  Act-Param [4], Meta-baseline[7]. To make the comparisons fair, we check and re-run the works with public code.
> - **ConvNet**  For IMP, the open-source code uses the val dataset as the test dataset, so we report our re-run results. No encoders are pre-trained for the ConvNet backbone. For CovaMNet, the source code uses very large sizes of validation episodes for model selection. We think in few-shot learning, a small validation data better aligns with the original intention, so we re-run the code in our setting.
> - **ResNet-12** Variational FSL is not open-sourced (and it is not easy to re-implement them), so we directly report their results in the paper. Note that, the variational fsl *pre-train* an encoder on the entire training data, and pre-training is also adapted in recent baselines like Meta-baseline. All the baselines use the same backbone, but the pre-training may be slightly different (some works are not open-source).
> - **WideResNet** This backbone is widely used in recent few-shot learning papers, all the methods in this block use the same backbone and pre-trained features.
>
> |Model|Backbone| 5-1 | 5-5 |
> |-|-| - |-|
> |IMP|ConvNet|33.30 ± 0.71|65.88 ± 0.71|
> |Prototypes |ConvNet|46.44 ± 0.60|63.72 ± 0.55|
> |CovaMNet|ConvNet| 51.83 ± 0.64|65.65 ± 0.77|
> |Big Prototypes|ConvNet|50.21 ± 0.31|66.48 ± 0.71|
> |SNAIL|ResNet-12|55.71 ± 0.99|68.88 ± 0.92|
> |Prototypes |ResNet-12|53.81 ± 0.23|75.68 ± 0.17|
> |Variational FSL |ResNet-12|61.23 ± 0.26|77.69 ± 0.17|
> |Prototypes + TRAML |ResNet-12|60.31 ± 0.48|77.94 ± 0.57|
> |Meta-baseline |ResNet-12|63.17 ± 0.23|79.26 ± 0.17|
> |Big Prototypes|ResNet-12|59.65 ± 0.62|78.24 ± 0.47|
> |Prototypes |WideResNet-28-10|59.09 ± 0.64|79.09 ± 0.46|
> |Activation to Parameter |WideResNet-28-10|59.60 ± 0.41|73.74 ± 0.19|
> |LEO|WideResNet-28-10|61.76 ± 0.08|77.59 ± 0.12|
> |Big Prototypes |WideResNet-28-10|**63.78 ± 0.63**|   **81.29 ± 0.46**|
>
>
> - For relation extraction, we add meta-learning methods including Proto-HATT [6] (a model with hybrid prototypical attention), Meta Net [9], SNAIL [10], Meta-GNN [11], MLMAN [12] (a method specifically designed for FewRel), Proto-ADV (a prototype-enhanced method with adversarial training), BERT-pair [13] (a strong BERT baseline for few-shot relation extraction). Proto-HATT and MLMAN are not model-agnostic. And we do not select methods that need additional related data like Wikipedia or Wikidata since FewRel is annotated based on Wikipedia and Wikidata. Details are reported in Appendix.
> - For Proto-HATT, we re-run the code (no 1-shot results are reported in the original paper) and report the results. For Meta Net, SNAIL, Meta-GNN, Proto-ADV, we re-run the code and find that the results are close to the papers. For MLMAN, we report the results in the paper.
> - The results of FewRel 1.0 are (also updated in the paper):
>
> |Model| 5-1 | 5-5 | 10-1 |10-5|
> |-|-| - | - |-|
> |Meta Net| 64.46 ± 0.54 | 80.57 ± 0.48 | 53.96 ± 0.56 |69.23 ± 0.52|
> |SNAIL |67.29 ± 0.26|79.40 ± 0.22|53.28 ± 0.27|68.33 ± 0.26|
> |GNN CNN |66.23 ± 0.75|81.28 ± 0.62|46.27 ± 0.80|64.02 ± 0.77|
> |GNN BERT|75.66 ± 0.18|89.06 ± 0.23|70.08 ± 0.48|76.93± 0.29|
> |Proto-HATT |76.30 ± 0.06|90.12 ± 0.04|64.13 ± 0.03|83.05 ± 0.05|
> |MLMAN |82.98 ± 0.20|92.66 ± 0.09|73.59 ± 0.26|87.29 ± 0.15|
> |Proto CNN|69.20 ± 0.20|84.79 ± 0.16|56.44 ± 0.22|75.55 ± 0.19|
> |Big Proto CNN (Ours)|66.05 ± 3.44|87.31 ± 0.93|56.74 ± 1.06|77.87 ± 2.60|
> |Proto BERT|80.68 ± 0.28|89.60 ± 0.09|71.48 ± 0.15|82.89 ± 0.11|
> |Big Proto BERT (Ours)|**84.34 ± 1.23**|   **93.42 ± 0.50**  | **77.24 ± 6.05**   | **88.71± 0.31**|
>
>
> - The results of FewRel 2.0 are (also updated in the paper, the order is 5-1, 5-5, 10-1, 10-5):
>
> |Model| 5-1 | 5-5 | 10-1 |10-5|
> |-|-| - | - |-|
> |Proto-ADV CNN |42.21 ± 0.09|58.71 ± 0.06|28.91 ± 0.10|44.35 ± 0.09|
> |Proto-ADV BERT |41.90 ± 0.44|54.74 ± 0.22|27.36 ± 0.50|37.40 ± 0.36|
> |BERT-pair |56.25 ± 0.40|67.44 ± 0.54|43.64 ± 0.46|53.17 ± 0.09|
> |Proto CNN|35.09 ± 0.10|49.37 ± 0.10|22.98 ± 0.05|35.22 ± 0.06|
> |Big Proto CNN (Ours)|36.41 ± 1.43|55.50 ± 1.42|22.11 ± 0.58|40.82 ± 2.50|
> |Proto BERT|40.12 ± 0.19|51.50 ± 0.29|26.45 ± 0.10|36.93 ± 0.01|
> |Big Proto BERT (Ours)|**59.03 ± 3.68** |  **74.85 ± 4.59** | **45.88 ± 7.43**  |  **61.61 ± 4.69**|
>
>
>
> - Instead of few-shot learning, hyperbolic prototypes focus more on enabling effective learning in low-dimensional output spaces and exploiting hierarchical relations amongst classes with or without privileged information about class labels.

---

### Official Review · Reviewer_6kXL · 2021-11-01

**Correctness:** 3
**Technical Novelty And Significance:** 3
**Empirical Novelty And Significance:** 3
**Recommendation:** 6
**Confidence:** 3

**Main Review:**

Strengths:
- The proposed big prototype model is a simple and easy to understand extension of the prototype model, which brings more adaptability to the latter. The simplicity of the model makes it easy to generalize to tasks that require a prototype model.
- The experimental section has demonstrated the effectiveness of big prototypes in three tasks across NLP and CV, which are few-shot named recognition, few-shot relation extraction, and few-shot image classification.
- The additional analysis of the change of hypersphere radius with sample density is helpful to understand the mechanism of big prototype model more intuitively.
- The writing is mostly clear and easy to follow.


Weaknesses：
- In the paper, the metric of two vectors is calculated by a L2 norm distance function. However, in some tasks using prototype models, cosine similarity metric considered more appropriate. So can the big prototype model be effectively applied in such a scenario? Is there anything that needs to be adjusted?
- The feature representation of all samples of a class often has a specific manifold distribution, not necessarily Gaussian distribution. What is the effect on the manifold distribution of feature representation before and after optimization with big prototype model? Could relevant visualization results or analysis be provided to show this impact?

**Summary Of The Paper:**

In few-shot learning, prototypes have been widely used to represent classes and then classification can be performed by computing distances to prototype representations of each class. This paper proposes to use hyperspheres to model prototypes in the feature space, instead of vector points, to enhance the expressivity of class-level information. The proposed hypersphere model only needs two sets of parameters (the center and the radius of hyperspheres), so it does not bring additional burden to the optimization calculation of the objective function. Extensive experiments in both NLP and CV are conducted to evaluate the effectiveness of the proposed model. The results shows that the proposed model significantly outperforms the baseline, and it performs well in cross-domain few-shot relation extraction.

**Summary Of The Review:**

As the author stated in Section 4, the experimental goal is not to lead in all the leaderboards, but to verify the effectiveness of the model only with baseline. Therefore, from this point of view, the proposed model is indeed better than the baseline model for comparison.

For me, the method proposed in the article is simple and has aroused my thinking. At the same time, it also brings some more worthy questions. I expect the author’s answer. At present, I think the proposed model is effective and enlightening. Therefore, I tend to suggest acceptance. Of course, I am also happy to see different opinions from other reviewers, and further adjust the score according to the feedback of the author.

---

> ### Author Response · Authors · 2021-11-17
> **Response to Reviewer 6kXL**
>
> Sincerely thanks for the comments, we believe they could help us improve the paper.
>
>
> ### Cosine and Euclidean metrics
>
> - We use the current Euclidean metric to align with our original motivation, that is, consider the radius $r$ of one hypersphere and make it participate in the learning stage. A cosine metric of two vectors is calculated by $d(x, y) = \frac{x\cdot y}{||x||||y||} = \cos \theta$, directly using this metric will violate our motivation and the current interpretability.
> - However, it is still an interesting question to ponder over, and the key point here is to introduce a parameter that could execise as a radius $r'$ just like in Euclidean metric. Assume that our points are all distributed on a unit ball, and a cosine metric $\cos \theta$ measures the angle between two vectors. If we set the newly introduced "radius" $r'$ as an angle (maybe using $\theta'$ is more appropriate), we will get a cone for a prototype (let's denote the center with $x$). $\theta'$ could be estimated by points from the support set.
> - Then the distance from a query point $y$ to a prototype is $\cos (\theta - \theta') = \frac{x'\cdot y}{||x'||||y||}$, where $x'$ is at the intersection of the arc of the cone on the sphere and the line between $x$ and $y$. Remember our goal here is to make $\theta'$ participate in the training, so we calculate $\theta = \arccos \frac{x\cdot y}{||x||||y||}$ and directly calculate $\cos (\theta - \theta')$.
> - In this way, we implement big prototypes with cosine metrics. Thanks for the inspiration, we will empirically try it out and see what will happen. If it is verified to be effective, we will add it to the final paper. Under this spirit, maybe more metrics could be introduced.
>
>
> ### Feature distribution
>
> - The analysis part reports the radius change according to the sampling in the learning phase. We agree that visualization of distributions will be better, so we add it in the Analysis section and the illustration shows that big prototypes could effectively learn discriminative features.
> - Besides, we also re-run previous baselines across different encoders to demonstrate the effectiveness of big prototypes.

---

> > ### Comment · Reviewer_6kXL · 2021-11-30
> > **About metric extension schemes**
> >
> > Thank you for your response.
> > The idea of extending distance metric to angle difference calculation is reasonable and feasible. I look forward to seeing more evaluation results in the final version. So far, my concerns have all been addressed, and I recommend accepting the paper.

---

### Official Review · Reviewer_bJMv · 2021-11-03

**Correctness:** 2
**Technical Novelty And Significance:** 1
**Empirical Novelty And Significance:** 1
**Recommendation:** 5
**Confidence:** 4

**Main Review:**

The paper extends and mainly compares with prototypical network published in 2017.
However, there are many works trying to improve prototypical network to model the prototypes more appropriately in literature. To name a few, see the following

[1] Distribution Consistency Based Covariance Metric Networks for Few-Shot Learning, AAAI-19
[2] Variational Few-Shot Learning, ICCV-19
[3] A Two-Stage Approach to Few-Shot Learning for Image Recognition, TIP-19

As for ``Ren et al., 2018; Gao et al., 2019a; Allen et al., 2019; Pan et al., 2019; Ding et al., 2021a" mentioned in introduction, can you compare with them empirically? Instead of just saying they "cannot uncover the overall ground truth distribution".
Why using hyperspheres  is better than using a distribution? Any theoretical analysis? Especially, please note that the hypersphere is still regular, while a distribution can be complicated.
Without fully discuss and empirically compare with the above-mentioned works, I cannot judge whether the performance gain is meaningful.


**Summary Of The Paper:**

The paper proposed to represent prototypes by hyperspheres with dynamic sizes.
A so-called big prototype is characterized by the center of the hypersphere and the radius of the sphere.
Empirical results are conducted on few-shot named entity recognition (NER), few-shot relation extraction (RE) and few-shot image classifica- tion.


**Summary Of The Review:**

The paper lacks  thorough literature review. To decide whether this solution indeed works, the authors should fully discuss and empirically compare with existing works. Just comparing with prototypical network is far from enough.

==== After rebuttal

Thanks for the new empirical results. I raise the score because the empirical part is now more completed.
However, it is still unknown why using hyperspheres can obtain better performance than others (especially those using distribution)  consistently. More insights and theoretical analysis are still required.

---

> ### Author Response · Authors · 2021-11-17
> **Response to Reviewer bJMv (2/2)**
>
> #### Discussion
>
> - We did not claim hyperspheres are "better" than distributions in our paper. But we agree that comparisons and discussions should be made. Empirically, big prototypes are easier to implement and achieve good performance in the last response. Also Euclidean metric is easier to implement than "point-to-distribution distance" [which could be measured by probability density function (maybe intractable) or the metric defined in CovaMNet]. And another advantage of our method is that it requires fewer parameters. We add a discussion part in Appendix in the revision.

---

> ### Author Response · Authors · 2021-11-17
> **Response to Reviewer bJMv (1/2)**
>
> Sincerely thanks for the comments.
>
>
> ### Compared with other methods
>
> - We agree that more baselines need to be compared and big prototypes could be adapted to more scenarios. Here are the empirical results.
> - For image classification, we add the following baselines, IMP (infinite mixture of prototypes) [1], CovaMNet [2], Variational FSL [3], LEO[5],  Act-Param [4], Meta-baseline[7]. To make the comparisons fair, we check and re-run the works with public code.
> - **ConvNet**  For IMP, the open-source code use the val dataset as the test dataset, so we report our re-run results. No encoders are pre-trained for ConvNet backbone. For CovaMNet, the source code uses very large sizes of validation episodes for model selection. We think in few-shot learning, a small validation data better aligns with the original intention, so we re-run the code in our setting.
> - **ResNet-12** Vairational FSL is not open-sourced (and it is not easy to re-implement them), so we directly report their results in the paper. Note that, the variational fsl *pre-train* an encoder on the entire training data, and pre-training is also adapted in recent baselines like Meta-baseline. All the baselines use the same backbone, but the pre-training may be slightly different (some works are not open-source).
> - **WideResNet** This backbone is widely used in recent few-shot learning papers, all the methods in this block use the same backbone and pre-trained features.
>
> |Model|Backbone| 5-1 | 5-5 |
> |-|-| - |-|
> |IMP|ConvNet|33.30 ± 0.71|65.88 ± 0.71|
> |Prototypes |ConvNet|46.44 ± 0.60|63.72 ± 0.55|
> |CovaMNet|ConvNet| 51.83 ± 0.64|65.65 ± 0.77|
> |Big Prototypes|ConvNet|50.21 ± 0.31|66.48 ± 0.71|
> |SNAIL|ResNet-12|55.71 ± 0.99|68.88 ± 0.92|
> |Prototypes |ResNet-12|53.81 ± 0.23|75.68 ± 0.17|
> |Variational FSL |ResNet-12|61.23 ± 0.26|77.69 ± 0.17|
> |Prototypes + TRAML |ResNet-12|60.31 ± 0.48|77.94 ± 0.57|
> |Meta-baseline |ResNet-12|63.17 ± 0.23|79.26 ± 0.17|
> |Big Prototypes|ResNet-12|59.65 ± 0.62|78.24 ± 0.47|
> |Prototypes |WideResNet-28-10|59.09 ± 0.64|79.09 ± 0.46|
> |Activation to Parameter |WideResNet-28-10|59.60 ± 0.41|73.74 ± 0.19|
> |LEO|WideResNet-28-10|61.76 ± 0.08|77.59 ± 0.12|
> |Big Prototypes |WideResNet-28-10|**63.78 ± 0.63**|   **81.29 ± 0.46**|
>
>
> - For relation extraction, we add meta-learning methods including Proto-HATT [6] (a model with hybrid prototypical attention), Meta Net [9], SNAIL [10], Meta-GNN [11], MLMAN [12] (a method specifically designed for FewRel), Proto-ADV (a prototype-enhanced method with adversarial training), BERT-pair [13] (a strong BERT baseline for few-shot relation exrtaction). Proto-HATT and MLMAN are not model-agnostic. And we do not select methods that need additional related data like Wikipedia or Wikidata since FewRel is annotated based on Wikipedia and Wikidata. Details are reported in Appendix.
> - For Proto-HATT, we re-run the code (no 1-shot results are reported in the original paper) and report the results. For Meta Net, SNAIL, Meta-GNN, Proto-ADV, we re-run the code and find that the results are close to the papers. For MLMAN, we report the results in the paper.
> - The results of FewRel 1.0 are (also updated in the paper):
>
> |Model| 5-1 | 5-5 | 10-1 |10-5|
> |-|-| - | - |-|
> |Meta Net| 64.46 ± 0.54 | 80.57 ± 0.48 | 53.96 ± 0.56 |69.23 ± 0.52|
> |SNAIL |67.29 ± 0.26|79.40 ± 0.22|53.28 ± 0.27|68.33 ± 0.26|
> |GNN CNN |66.23 ± 0.75|81.28 ± 0.62|46.27 ± 0.80|64.02 ± 0.77|
> |GNN BERT|75.66 ± 0.18|89.06 ± 0.23|70.08 ± 0.48|76.93± 0.29|
> |Proto-HATT |76.30 ± 0.06|90.12 ± 0.04|64.13 ± 0.03|83.05 ± 0.05|
> |MLMAN |82.98 ± 0.20|92.66 ± 0.09|73.59 ± 0.26|87.29 ± 0.15|
> |Proto CNN|69.20 ± 0.20|84.79 ± 0.16|56.44 ± 0.22|75.55 ± 0.19|
> |Big Proto CNN (Ours)|66.05 ± 3.44|87.31 ± 0.93|56.74 ± 1.06|77.87 ± 2.60|
> |Proto BERT|80.68 ± 0.28|89.60 ± 0.09|71.48 ± 0.15|82.89 ± 0.11|
> |Big Proto BERT (Ours)|**84.34 ± 1.23**|   **93.42 ± 0.50**  | **77.24 ± 6.05**   | **88.71± 0.31**|
>
>
> - The results of FewRel 2.0 are (also updated in the paper, the order is 5-1, 5-5, 10-1, 10-5):
>
> |Model| 5-1 | 5-5 | 10-1 |10-5|
> |-|-| - | - |-|
> |Proto-ADV CNN |42.21 ± 0.09|58.71 ± 0.06|28.91 ± 0.10|44.35 ± 0.09|
> |Proto-ADV BERT |41.90 ± 0.44|54.74 ± 0.22|27.36 ± 0.50|37.40 ± 0.36|
> |BERT-pair |56.25 ± 0.40|67.44 ± 0.54|43.64 ± 0.46|53.17 ± 0.09|
> |Proto CNN|35.09 ± 0.10|49.37 ± 0.10|22.98 ± 0.05|35.22 ± 0.06|
> |Big Proto CNN (Ours)|36.41 ± 1.43|55.50 ± 1.42|22.11 ± 0.58|40.82 ± 2.50|
> |Proto BERT|40.12 ± 0.19|51.50 ± 0.29|26.45 ± 0.10|36.93 ± 0.01|
> |Big Proto BERT (Ours)|**59.03 ± 3.68** |  **74.85 ± 4.59** | **45.88 ± 7.43**  |  **61.61 ± 4.69**|

---

> ### Author Response · Authors · 2021-11-29
> **Response to Reviewer bJMv (3)**
>
> Thanks, we will add more analysis and details in the next version of this paper

---

### Official Review · Reviewer_4NsC · 2021-11-03

**Correctness:** 4
**Technical Novelty And Significance:** 3
**Empirical Novelty And Significance:** 3
**Recommendation:** 6
**Confidence:** 2

**Main Review:**

Strengths:
1) The motivation of the paper is interesting. Using two sets of learnable parameters to represent the class prototype can reduce the influence of biased data.
2)The metrics of the proposed big prototypes are easy to calculate. Both the radius and hyper-spheres center participate in the backward propagation, which leads to the adjustable centers and radius corresponding to different categories.
3) Extensive experiments in both NLP and CV evaluate the effectiveness of the proposed method, especially under the setting of cross-domian FSL learning.


Weakness:
1) More analysis on the experimental details are recommended. I am interested in the feature distribution in the training process. It seems that only the points outside the hyperspheres affect the loss function. That means the proposed method can reduce the number of outliers of classes, and the learned features gather more closer in each class compared with the original prototypes method. It is better to provide some visualization of feature distribution.

2) The applicability of the method is somewhat limited. The result of 1-shot task on miniImageNet is missing. In 1-shot learning, the proposed model cannot estimate the radius of the class by a single image, so the model sill suffers from the point-biased limitation?



**Summary Of The Paper:**

The paper proposes to use areas to model prototypes in FSL. The prototypes, named as big prototypes, are represented by hyper-spheres with dynamic sizes. Rather than point-based prototypes, the new area-based prototypes  in the embedding space can represent the class-level information with more expressivity.

**Summary Of The Review:**

The paper proposes "big prototypes" for few shot learning, which represents the prototypes by hyper-spheres with dynamic sizes. The new method is easy to implement and can reduce the influence of biased data. Extensive experiments in both NLP and CV demonstrate the effectiveness of the proposed method. More details or analysis are needed to improve the paper.

---

> ### Author Response · Authors · 2021-11-17
> **Response to Reviewer 4NsC**
>
> Sincerely thanks for the valuable comments.
>
>
> ### More experimental details and analysis
>
> - We have uploaded a revision with many more baselines introduced to further demonstrate the effectiveness of big prototypes. All the experimental details are reported in Appendix.
> - And we agree that the illutrastions of feature distributions will be better. We add a visualization of distributions in the Analysis section. The visualization demonstrates the effectiveness of bigprototypes training in learning discriminative features.
>
>
> ### Only the points outside the hyperspheres affect the loss function?
> - In fact, all the points will participate and affect the loss function. According to our equation $d(x,y)=||x-y||^2_2 -d$, if a points is inside the hypersphere, the corresponding metric will be minus, which will not affect the calculation of the exponential function. In fact, we find that this strategy aligns with our intuition and pushes more points "inside" the hypersphere and also optimizes the parameters of big prototypes.
>
>
> ### 1-shot experiments
> - In the 1-shot setting, the big prototypes will still help the training. Since it will be hard to estimate the radius because we only have one point for each class, we can set the initial radius as a hyper-parameter and optimize it during training.
> - The radius could still ensure the intra-class compactness and inter-class separability in the training. And 1-shot results on FewRel and miniImageNet could demonstrate the effectiveness of big prototypes.
> - In the last version of our paper, we did not report 1-shot results for image classification because the k-shot (k>1) setting is more in line with the motivation of the big prototypes and makes it easier to interpret the performance. Now we add the 1-shot results in the revised paper to make better comparisons with previous approaches.

---

### Author Response · Authors · 2021-11-17
**General Response to All the Reviewers**

We thank all the reviewers for your valuable comments and upload a revised paper. We have made the following modifications.
- More baselines according to the suggestions of Reviewer bJMv, 7SK7 and 4NsC. We demonstrate what we have done to make the comparisons fair in our response.
- Polish the paper to clarify some confusing details mainly according to the comment of Reviewer 7SK7.
- Add a discussion about our method and other prototype-enhanced models in Appendix according to the comment of Reviewer bJMv.
- Add visualization according to the comments of Reviewer 4NsC and 6kXL.
- Move experimental details to Appendix.

We list references of baselines here:
- [1] Infinite Mixture Prototypes for Few-Shot Learning. ICML 2019.
- [2] Distribution Consistency Based Covariance Metric Networks for Few-Shot Learning. AAAI 2019.
- [3] Variational Few-Shot Learning. ICCV 2019.
- [4] Few-shot image recognition by predicting parameters from activations. CVPR 2018.
- [5] Meta-learning with latent embedding optimization. ICLR 2019.
- [6] Hybrid attention-based prototypical networks for noisy few-shot relation classification. AAAI 2019.
- [7] Meta-Baseline: Exploring Simple Meta-Learning for Few-Shot Learning. ICCV 2021.
- [8] Boosting few-shot learning with adaptive margin loss. CVPR 2020.
- [9] Meta Networks. ICML 2017.
- [10] A Simple Neural Attentive Meta Learner. ICLR 2018.
- [11] Few-shot learning with graph neural networks.ICLR 2018.
- [12] Multi-Level Matching and Aggregation Network for Few-Shot Relation Classification. ACL 2019.
- [13] FewRel 2.0: Towards More Challenging Few-Shot Relation Classification. EMNLP 2019.

---

### Decision · Program_Chairs · 2022-01-20

**Decision:**

Reject

**Comment:**

Thanks for your submission to ICLR.

This paper presents an extension to prototypical networks based on using hyperspheres to represent the prototypes.  Strong empirical results are presented using this approach.

Overall, this is a very borderline paper and could go either way.  The idea itself it simple, though the results seem to be fairly strong.  I read through the paper myself and tend to think that it could use a bit more work before it's ready.  Some of the issues raised by the reviewers---particularly with respect to experiments and literature review---are worth nailing down.  Further, I think that the method could be explored in a more principled/theoretical way.  For instance, when reading this idea, the first thing that pops into my mind is that representing the prototype with a hypersphere is very similar to representing a distribution (e.g., a Gaussian) using a mean and covariance (in this case, a spherical covariance).  Indeed, if you take the KL divergence between two spherical Gaussians, you get something very similar to the expression used in the paper.  This is all to say that there may be other more general directions to take this idea, or other interpretations of what is going on.

Please do keep in mind the comments of the reviewers when preparing a future version of the manuscript.